# "Refugees in the Amphitheatre": An Intercultural Action Research on Co-Educating Student Teachers and Peer Refugees

## Kostas Magos

Department of Early Childhood Education, University of Thessaly, 382 21 Volos, Greece; magos@uth.gr

**Abstract:** The contribution of action research to teacher education as well as to refugee education has been highlighted in the international literature. Through action research, teachers can link educational theories with everyday school practices. In addition, the participation of refugees in action research, especially in cooperation with members of the dominant ethnic and cultural group, could play a significant role in their empowerment and social inclusion. This article describes the content and the results of an action research, which took place in the context of an academic course in a Greek University. The aim of the action research was the interaction between students and peer refugees and, through it, the development of intercultural competence and empathy. The action research developed in three cycles, featuring the students and refugees' participation in intercultural routes–walks. The action research findings showed that the participation in the abovementioned walks supported the intercultural communication and interaction among the group members, as well as the reflection on refugee identity stereotypes.

**Keywords:** intercultural education; action research; refugees

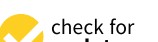



## 1. Introduction

The effectiveness of action research in both teacher education and refugee education has been repeatedly highlighted in the international literature. As far as teacher education is concerned, action research promotes experimentation and discussion on educational theories, so that teachers themselves become capable of articulating and documenting the teaching methods they use [1]. As ref. [2] argues, the above process contributes not only to the teachers' personal growth and development, but also to leading towards a more general, critical approach and, ultimately, to the improvement of the school institution, which, as ref. [3] points out, gradually needs to be transformed from a typically operational mechanism into a dynamic organism that learns and evolves. In particular, for prospective and new teachers, action research can be the right environment for them, to develop into agents of innovative educational and social interventions [4], but also to understand the importance of the research process in the context of everyday school practice.

According to ref. [5], in-service teachers usually avoid getting involved with the research process and, specifically, with action research, because they consider that they have not acquired sufficient knowledge and experience in their studies regarding the implementation of action research in the context of their work. Therefore, the participation of prospective teachers in the planning and implementation of action research works positively toward their adopting the role of teacher–researcher in their future professional development.

Accordingly, action research can provide important benefits as regards issues related to refugee education. Ref. [6] describes how the participation of young refugees in an action research, which took place under particularly difficult conditions in a refugee camp in Kenya, contributed to the empowerment of the participants, the mobilization of their critical thinking and the cultivation of their empathy. Ref [6] underlines that the participatory action research—by promoting decision-making and, therefore, active participation of research subjects—can contribute to the creation of truly collaborative relationships. In

many cases, refugees refuse to cooperate with the researchers in the context of traditional researches, believing that any such research findings do not contribute to the improvement of their own lives, whereas they seem to be more willing to participate in action researches.

The participation of prospective teachers and their refugee peers in a common action research not only works positively for each group, but at the same time creates suitable conditions for the development of intercultural competence and empathy. Cultural and other stereotypes often inform the perceptions of members of each of the above groups; so, the members' meeting and collaboration during an action research can contribute not only towards overcoming stereotypes, but also towards developing intercultural communication and exchange.

According to ref. [7], educators' negative perceptions and attitudes towards various dimensions of refugee education continue to exist, although they seem to have decreased compared to the past. The authors argue that such attitudes seem to be linked to the educators' general fear of a subsequent change of what is considered to be the dominant ethnic group's established privileges but, also, to the low rates of teacher specialization and training regarding issues arising in intercultural education.

Such negative perceptions could probably be transformed through the teachers' acquaintance and interaction with the refugees. The joint participation of prospective teachers and refugees in an action research can create a suitable environment for the development of intercultural interaction.

Ref. [8] uses the term "intercultural action research" for the action research that prioritizes the development of the participants' intercultural competence and empathy. Ref. [9] point out how engaging teachers in action research with an intercultural purpose helped practicing teachers to transcend ethno-cultural stereotypes and understand the important role that they themselves play in the formation of cultural identities. To this effect, ref. [10] highlight the benefits of teachers' participation in intercultural action researches, since through them, they can gain a deeper understanding of educational and social inequalities and support educational processes that promote equal opportunities for all students. In addition, ref. [11] highlights that the participation of candidate teachers, together with young unaccompanied asylum seekers, in an action research, worked positively towards the future teachers' personal growth and development of intercultural competence.

## 2. Materials and Methods

The action research presented here was planned and carried out in the context of the course "Planning, organization and evaluation of intercultural actions", which is taught in the Department of Preschool Education, at the University of Thessaly in Greece, with the writer as educator. The purpose of the course is the development of students' intercultural competence and empathy through their participation in educational and social actions with intercultural content. In recent years, the course has focused on the development of participants' empathy towards the refugee experience; and for this purpose, there is systematic cooperation with a refugee accommodation center, run by an NGO in the same town as the university. The students that participate in the course co-decide with the educator on the content of the action research, following relevant proposals put forward by the educator.

The acquaintance of the students with the process of action research and especially with the purpose of an "intercultural action research" has already taken place in a previous semester of their studies. Moreover, the curriculum of the above university department includes other courses that focus on intercultural and anti-racist education; but this particular course is the only one that utilizes experiential actions, in order to develop intercultural interaction between students and members of minorities and other non-dominant groups. The educator's basic position is that cultural interaction, in the context of participation in common actions, can lead to the development of participants' intercultural competence and empathy.

In the context of the above course, in the spring-semester 2018–19, it was decided that the central theme of the action research would be the mutual acquaintance and interaction between the students and peer refugees who stayed in the aforementioned accommodation center. The purpose of the research was to investigate the interaction between students and peer refugees in the context of co-education during an academic course using experiential actions. Given the limited number of researches and educational implementations that make use of the co-education of refugees and native learners, this specific research becomes particularly important and can contribute to the more general literature concerning the education and inclusion of learners with refugee experience.

Especially in the case of Greece, where, due to its geographical location, refugee flows have been particularly high in the last decade, and the presence of refugee students both in the context of formal and non-formal education is significant, the above research can highlight interesting findings on the educational and social inclusion of refugee learners. At the same time, it can highlight effective methods of formal and non-formal education of the native population, in order to reduce stereotypes and prejudices against refugees. It is worth mentioning that stereotypes and prejudices against both refugees and immigrants, although reduced compared to the past, continue to appear in the opinions and attitudes of members of the dominant ethno-cultural group [7].

The key questions of the action research presented here were formed by the students and were related to the refugees' daily life, their relations with the inhabitants of the city, as well as to the likelihood of discriminations against refugees and/or racist attitudes towards them. Of particular interest is the way the students themselves formulated the above questions in the context of a relevant "brainstorming session". Thus, the investigation of the refugees' lives and daily reality was formulated with the question: "Where do these 'hidden' people live?"—highlighting the absence of any communication, even visual, between the refugees and the local population. Similarly, the desire to investigate possible discrimination suffered by the refugees was formulated with the question: "Have they been 'kicked' while living here?"—a question that partly reveals the students' suspicion of a negative attitude on the part of the dominant population towards this particular group.

In tandem with the above key questions, the students formulated questions concerning individual issues, related to the life of the refugees, such as issues of work and unemployment, language, communication and education, entertainment and future goals. Generally, the students—most of whom came from small provincial places in Central and Northern Greece—presented the absence of personal reflection on this particular group hitherto and the replication of perceptions about the refugees through the most popular mass media. None of the students had met or communicated with a refugee until the time of their participation in the action research, whereas they had met immigrants.

Taking into account the above information, the educator proposed to the ten-member group of students, who would participate in the action research, that a group of peer refugees from the local accommodation center would be invited, in order to participate in the course. The proposal was accepted by both sides and thus, a twenty-member group of ten students and ten young refugees originating from Pakistan, Afghanistan, Bangladesh, Iraq and Syria was created. The course instructor had the role of the facilitator-coordinator of the action research, while the role of a critical friend was covered by a researcher, working in the refugees' accommodation center.

The participation of refugees in educational and community actions, together with non-refugees, is highlighted as one of the most basic dimensions for the educational and social integration of the refugee population [12,13]. The meeting of refugees with members of the local community contributes to a better and faster approach to the other culture, creates communication links and promotes an integration framework both in matters of education and in general.

At the first meeting of the group of students and refugee peers, which took place in the university auditorium, the action research was presented as a project to get to know each other and develop intercultural communication. There were no significant language

communication problems despite the different mother tongues of the participants, as some of the refugees already knew Greek at a satisfactory level and they themselves acted as interpreters for those who did not.

At a separate time, the use of a research diary was discussed and agreed upon by the students. The main purpose of the diary was to record, on the one hand, the information they would acquire in relation to the initial questions they themselves had asked about the life and daily routine of the refugees and, on the other, their thoughts and feelings, as they would be shaped through their meetings with the rest of the team members. A research journal would also be kept by both the educator and the critical friend.

In addition to the questions that the participating students wanted to investigate in the context of the action research, an additional basic research question emerged for the course educator. This concerned whether and to what extent the participation of students in the specific project could transform previous perceptions and attitudes towards refugee status. In other words, if the specific experience could function as a disorienting dilemma, according to the term used by ref. [14], capable of leading the participating students through reflection to the creation of new mental perceptions, as well as attitudes, towards refugees.

In order to explore the above question, the educator, using the focus group interview technique, recorded the perceptions of the participating students towards refugees and refugee status before starting the action research. The dominant perception appeared to be that of compassion, a perception quite widespread in general, but not compatible with the concept of intercultural competence and empathy, since the attitudes of compassion and pity indirectly signal cultural hierarchy towards the cultural 'other'. This perception was explicitly expressed by a small minority of the student group. The students speculated that most of the refugees do not have an education and that, due to the conditions they live in, they are likely to develop delinquent behavior (if they do not have food, they will steal/since there is a war in their country, what school should they go to . . . ).

During the action research, four different categories of meetings took place. The first included the meetings of all participants (educator, students, refugees, critical friend). A total of ten such meetings took place, the description of which follows. The second category included the meetings of the smaller groups (students, refugees) which met in order to prepare the material and individual topics of the meetings of the previous category. The third category included the meetings between the educator-coordinator of the action research and the students. These meetings took place at the University and included reflective discussions, research feedback, utilization of research diaries and a general connection between action and research in order to make the context of action research as clear as possible. Finally, the last group of meetings was the one between the educator-coordinator and the critical friend, with the purpose of discussing the development of the action research, feedback, anticipation and overcoming of possible difficulties.

## 3. Results

### 3.1. First Cycle of Action Research

The first cycle of the action research included two meetings among all involved in order to get to know each other. In these meetings, activities from the techniques of "breaking the ice" were utilized, as well as the storytelling technique, i.e. narratives, in small groups of participants, about experiences and incidents of everyday life. Narratives are not only an effective method of getting to know each other, since through them the subjects exchange opinions and experiences, but also an important means of reflection, as it helps the narrators to 'see outside the box' [15].

Ref. [16] underlines the importance of refugees' narratives, as they act as a mirror through which they can both 'see themselves' and be seen by others. Ref. [17] talks about critical narratives, i.e., those narratives that promote critical thinking and reflection on more general issues, such as social inequalities and discrimination. Particularly important are the so-called "covered narratives" [18], i.e., the narratives of members of vulnerable groups, who do not directly express what they want to say, mainly for fear of the consequences they

may suffer from members of dominant groups. Narratives of refugees, especially those that focus on discrimination and racist attitudes they may have suffered, belong to the covered narratives.

The two initial meetings seem to have sparked the interest of both the students and their refugee peers to such an extent that they kept on having more meetings, with each group acquiring more information about the other group. The reflective meetings that took place with the students in the context of the course focused on the characteristics of the refugee identity. The image of the refugees ideated by the majority of the students—obviously influenced by corresponding media images—was very different from the image of the young refugees they met in the context of the action. As F. characteristically said, "I expected to meet poorly dressed, hungry, ugly people and not better-than-ours mobile phones and fashionable clothes." Accordingly, K. found that "A.'s English was better than mine and he told me that he spoke three more Asian languages." The subversion of a constructed archetypal image of the refugee and the finding that refugee identity constitutes an element of the individuals' personality, important, but not unique, emerged as the dominant observation by the students in the reflective discussion that followed the first two meetings.

To this effect, and according to the entries in the critical friend's research diary, the refugees' observations focused on the immediacy of communication that had developed despite the different language, especially with people of the opposite sex since, in their countries of origin, intimate relations between non-relatives of different sexes were forbidden.

In the second reflective meeting, which also included the planning of the next actions of the action research, one of the members of the student group suggested that the meetings of students and refugees should take place in third places, preferably out of doors, outside the university amphitheater. The student documented this desire by arguing that the meeting of the group members in different places would give new, interesting stimuli to promote mutual acquaintance among them. This view was discussed in depth by the educator and the critical friend, eventually, ending up in proposing a next round of action research. In this cycle, the meetings of the group members would take the form of five routes–walks in the city and in the wider area with specific objectives and corresponding stops.

*3.2. Second Cycle of Action Research*

The routes planned in the context of the second cycle of the action research included getting to know the 'hangouts' of the students on the one hand and the refugees on the other, routes to the old town, 'literary' walks and natural routes to the suburbs. For each route, the smaller groups of students and refugees prepared suitable material and related activities, while in collaboration with the educator-coordinator they decided the stops of each route, as well as the main topics of discussion at each stop.

For example, the route to the Old Town, with the ruins of an Ottoman mosque as well as an Ottoman inscription, gave rise to discussing issues related to differences and similarities between religions, but also to the phenomenon of religious fundamentalism and its consequences both in the refugees' countries of origin as well as in Europe. Similarly, the "literary walk" was a cause for the young refugees to get to know important personalities of the Greek literary tradition and for the students, thanks to the refugees, to be informed of important poets and writers originating from the refugees' countries of origin. As A. typically commented at the feedback meeting, "Twelve years of school and I had never heard of a writer or poet from the countries where the refugees come. As if these countries only have poverty and war, nothing else."

Both the facilitator and the critical friends of the action research highlight in the excerpts of their research diaries the effectiveness of the walks on achieving the goal of cultivating intercultural communication and exchange. Key advantages of this "peripatetic action research" were that the peripatetic context appeared to be a particularly suitable context for cultivating collaboration and communication, as it enabled team members to approach each other more easily, respected the individual rhythms of each member and

stressed the element of pleasure as a key element of the learning process. Obviously, the above process has the characteristics and advantages of a non-formal education, where the experiential element of learning is predominant, while the approach to knowledge is made as a result of discussion and cooperation among all members of the group.

As the meetings developed in the framework of the peripatetic action research, the development of communication and interaction between the students participating in the action and the peer refugees became more and more evident both in life and through the diary entries. The following entry from student L's diary is typical. "Today H. told me about his life in Pakistan. [...] After all that I learned about X., I feel him a bit like my cousin. We'll stay in touch, even if he leaves for Germany soon, as he thinks." To that effect, the critical friend recorded in his diary that the young refugees preferred to lose their daily wages, which they hardly secured, if their working day coincided with a walking day. From this observation, it becomes clear that, despite the very difficult financial situation they were in, they prioritized participation in the group and in walking activities as more important. This opinion was confirmed by them in a subsequent meeting, when they expressed the desire for the meetings to have a longer duration than the four hours they had until then. The educator suggested to all the group members to decide about the duration and content of the next meetings. The final proposal resulted in the common desire to organize two all-day meetings, in the form of a day trip, once every month for the next two following months. The proposal was accepted and essentially formed the starting point of a third cycle of action research.

*3.3. Third Cycle of Action Research*

Although the routes and the relevant stops were decided by the educator-coordinator and critical friend, a large part of the preparation was in terms of the material that was to be used, the activities that were to be carried out and also the food which was prepared by the smaller groups of students and refugees. One of the two meetings had as a main stop a point connected with the mythology associated with the area and related to the cave of the Centaur Chiron, while the second focused on a mountainous regional route using a local train.

In the context of the first meeting, prompted by the myth of the Centaur Chiron, the young refugees were given the opportunity to present heroes from the mythology of their places of origin, looking for similarities and differences with those of the Greek mythology. Coming across an old Ottoman fountain at a way station gave new food for thought and discussion about fairies and elves, fairy-tale elements that seem to run through the folk traditions of the countries of origin of all the young refugees.

The short trip, which was made as part of the second all-day meeting, was a route in a mountainous area. It reminded many of the young refugees of the usual ways of traveling in their own places of origin, while, at the same time, it stimulated them to talk both about them (the ways of traveling) and about their journey to Greece, since, apart from the other means of transport, they had also used the train. Yet, as they typically said: 'not the inside train, but the outside train', that is, the roof of the train, on which they were traveling. Given that the second all-day meeting also marked the end of the walking tours, sufficient time was dedicated to the evaluation of the whole action by all the members of the group, prior to its end. Sitting in a circle, all the members of the group gradually brought to their memory one by one the meetings they participated in. Both students and peer refugees focused on the mutual acquaintance and mutual communication achieved and the desire to maintain the relationship created. As the critical friend wrote in his diary, in the above conversation, the phrase that was heard most often and from many different mouths was: "I made new friends".

One might interpret the above sentence as a platitude. However, in this last meeting, what the group of students and peer refugees showcased—complete strangers to each other only a few months ago, with different native languages but who communicated,

laughed, sang, and were jointly organizing a subsequent meeting of their own—was that, to a significant extent, the development of intercultural communication had been achieved.

## 4. Discussion

In the final reflective meeting of the course between the educator and the students, making use of the focus group interview again, the development of empathy towards the refugees was confirmed. All the students agreed, to different degrees, that their participation in the specific project was very helpful for them in understanding the refugee experience in depth, reflecting on issues of discrimination and social inequalities, but also getting to know cultural characteristics of people and cultures, unknown to them prior to their participation in the specific action.

Although the element of action was much more intense than that of research, a fact that was also observed and analyzed in a previous relative action research [8], the majority of the participating students seem to have understood to a significant extent the context, function and action research methodology. Moreover, they have learnt how they could keep a research diary, excerpts from which they read in the reflective meetings. The fact that two of them subsequently decided to choose this particular project as the subject of their final paper was an additional way of confirming the degree of their interest and involvement in the action.

Likewise, according to the evaluation of the critical friend who was also researcher in the refugee accommodation center, the participation of the young refugees in the whole process mobilized three of them to register in an evening high school. At the same time, as he pointed out in his evaluation report, the presence of the young refugees in the walking activities and the interaction with the students played a particularly important role in improving their mental health, developing self-confidence and improving their self-image. Given that in refugee accommodation centers there is usually no possibility of communication, moreover, of creating relationships with non-refugees, their participation in this action helped not only to meet peers of the dominant group, but also to have the opportunity to discuss, co-exist and exchange cultural knowledge with them for a long time. This communication contributed to a lesser or greater degree to their understanding that any discrimination and racist attitudes they may have received cannot mark the entire population of the host country, but only the specific individuals who carry particular racist perceptions and attitudes. In addition, their participation in this specific action contributed to better acquaintance and communication between them, as well as to the improvement of relations between them, regardless of their country of origin.

## 5. Conclusions

The action research presented above aimed to explore the interaction between students and peer refugees. As established by the findings of the research, through their participation in the action research, the students and the peer refugees had the opportunity to get to know each other and interact through experiential processes, such as the organized common routes–walks in different parts of the city where they live as well as of the wider area. In this context, the routes, in which the students and their peer refugees met and communicated, simultaneously were cognitive and transformative routes, which helped the participants to reflect on stereotypes and develop intercultural competence and empathy, elements that are directly connected with the targets of an intercultural action research.

Despite the fact that the findings of an action research cannot be generalized, since each action research is also a separate case study research [19], they are still important, since they can provide information that will be used in the design of subsequent research. Moreover, despite the limited number of participants, which could be a limitation of the research, this research showed that the experiential teaching and learning approach is a suitable approach for the development of intercultural interaction which can also lead to the development of empathy [20] in the context of co-education of students and fellow refugees. Similar action researches at other educational levels of typical and/or non-typical

education, such as elementary, middle and high-school level, as well as in adult education programs, can confirm the findings of the research presented above, as well as highlight other possible factors and educational approaches that may play a decisive role in the co-education of refugees and native learners.

**Funding:** This research received no external funding.

**Institutional Review Board Statement:** The study was conducted in accordance with the Declaration of Helsinki, and approved by the Scientific Committee for Refugees' Education and Inclusion in Volos/Greece Hospitality Centre for studies involving humans. The code is 52 and the date is 24 January 2018.

**Informed Consent Statement:** Informed consent was obtained from all subjects involved in the study.

**Data Availability Statement:** Not applicable.

**Conflicts of Interest:** The author declares no conflict of interest.

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
