# Peer review of "“Refugees in the Amphitheatre”: An Intercultural Action Research on Co-Educating Student Teachers and Peer Refugees"

_societies, doi:10.3390/soc13030060_

Round 1

Reviewer 1 Report

After careful scrutinizing of the content, it has been observed that the paper is good but has a lot of scope of further improvement in the following ways-

1.Grammatical Errors found in abstract and the rest of the content. Authors are required to revisit the whole paper and check the grammatical errors.

2.In the Introduction section, although authors have described the importance of intercultural action research in the context of Refugee Students, but objective of the present research is not well defined. Therefore, In the Introduction section, a small paragraph must be added by the authors indicating –i.the objective of the present research, ii. how the research is unique and contributing to the knowledge reservoir  .

3.Authors are required to restructure the paper in terms of the various sections of the research as for example Discussions and conclusion should be different headings.

4. Authors are advised to re frame the conclusion section to add more clarity and quality to the content of this research work.

Author Response

Dear reviewer,

thank you very much for your comments. I have done all the changes you proposed.

  1. The manuscript was checked for grammatical and other errrors by two English native speakers.
  2. I add paragraphs focusing on the aim of the research and its contribution to the knowledge reservoir. (lines 94-118).
  3. Discussion and conclusion are now different headings.
  4. I wrote two new paragraphs for the conclusions sections (lines 351-371)

Reviewer 2 Report

This is an interesting article that is current and compelling given the high number of refugees worldwide, including those entering Greece.

The author could contextualize this study more thoroughly to make this strong an interesting piece even more significant.  Specifically, I suggest adding in a few paragraphs in a background section to discuss the nature of refugee arrivals and their lifestyle in Greece.  There is a reference to folks essentially living in the shadows and the realities of trying to fit in, but readers without background knowledge of the high number of refugees in Greece, and the debt crisis, will miss out on the importance of their presence and student teachers' attempts to work with them.

Additionally, it would be good to know a bit more about what else the student teachers' curriculum includes and how this class/training fit in, even if that is in a footnote.

This should be a relatively easy modification for the author to make, and it will increase the quality and strength of the article.

Author Response

Dear reviewer,

thank you very much for your very interesting comments. I followed them and did the appropriate changes. I added some basic information concerning the refugees' condition in Greece as well  as the curriculum of the university courses they have participated (lines 93-117).

Round 2

Reviewer 1 Report

Dear Authors,

The revised version of your research work is ok now. But in reference section, all the reference should be placed in an alphabetical order and accordingly the reference numbering  must be corrected in the text of the manuscript also.